# A Causal View of Entity Bias in (Large) Language Models

**Fei Wang**[†] **Wenjie Mo**[†] **Yiwei Wang**[‡] **Wenxuan Zhou**[†] **Muhao Chen**[†♯]

[†]University of Southern California; [‡]University of California, Los Angeles;
[♯]University of California, Davis

{fwang598,jackymo,zhouwenx}@usc.edu; wangyw.evan@gmail.com;
muhchen@ucdavis.edu

## Abstract

Entity bias widely affects pretrained (large) language models, causing them to rely on (biased) parametric knowledge to make unfaithful predictions. Although causality-inspired methods have shown great potential to mitigate entity bias, it is hard to precisely estimate the parameters of underlying causal models in practice. The rise of black-box LLMs also makes the situation even worse, because of their inaccessible parameters and uncalibrated logits. To address these problems, we propose a specific structured causal model (SCM) whose parameters are comparatively easier to estimate. Building upon this SCM, we propose causal intervention techniques to mitigate entity bias for both white-box and black-box settings. The proposed causal intervention perturbs the original entity with neighboring entities. This intervention reduces specific biasing information pertaining to the original entity while still preserving sufficient semantic information from similar entities. Under the white-box setting, our training-time intervention improves OOD performance of PLMs on relation extraction (RE) and machine reading comprehension (MRC) by 5.7 points and by 9.1 points, respectively. Under the black-box setting, our in-context intervention effectively reduces the entity-based knowledge conflicts of GPT-3.5, achieving up to 20.5 points of improvement of exact match accuracy on MRC and up to 17.6 points of reduction in memorization ratio on RE.[1]

## 1 Introduction

Entity bias ([Longpre et al., 2021](#); [Wang et al., 2022](#); [Xu et al., 2022](#); [Peng et al., 2020](#); [Qian et al., 2021b](#); [Hermann et al., 2015](#)) refers to an undesirable phenomenon where models overly rely on prediction shortcuts triggered by specific entities to make spurious predictions. For example, given the sentence *"Bill Gates went to Microsoft Building 99,"* models

Figure 1: An example of entity bias in GPT-3.5. Our in-context intervention mitigates the conflicts between parametric knowledge and contextual knowledge.

may be misled by their memory of the entities *Bill Gates* and *Microsoft*, saying the relation between them in this context is *founder* rather than *visitor*, as shown in [Fig. 1](#). Recent studies show that entity bias widely affects pretrained (large) language models (LLMs; [Longpre et al. 2021](#); [Yan et al. 2022](#); [Zhou et al. 2023](#)). These models have a tendency to disregard contextual information that contradicts or is infrequently reported in the pretrained corpus, while excessively relying on (biased) parametric knowledge ([Longpre et al., 2021](#)) to make unfaithful predictions and perpetuate bias.

Prior studies have proposed multiple causality-inspired methods to mitigate entity bias ([Zhang et al., 2017](#); [Nan et al., 2021](#); [Wang et al., 2022](#); [Zhu et al., 2022](#)).[2] Despite their potential, the causal models underlying these methods are flawed in practice, primarily because of imprecise parameter estimation. For example, some causal models necessitate estimating the probability distribution

---

[2]Although [Zhang et al. (2017)](#) do not mention causal theory, the proposed entity masking does follow a relevant principle to cut off causal links between specific entities and labels.

over labels when given a sentence that is devoid of entities or contextual information (Zhang et al., 2017; Wang et al., 2022). These methods either lose predictive information about entities, or are prone to erroneous representation without contextualization. The other critical problem is the difficulty of applying these methods to black-box LLMs, of which parameters are inaccessible and logits are uncalibrated.

To address the aforementioned problems, the *first* contribution of this paper is a **causal analysis of entity bias mitigation methods** (§3.1). We examine and compare the structured causal models (SCMs) behind existing methods. We find that, among the theoretically equivalent causal models (Verma and Pearl, 1990), there exists a specific SCM whose parameters are comparatively easier to estimate. As shown in Fig. 2, the proposed SCM only requires to intervene input entities to mitigate the presence of spurious features before passing them to the subsequent neural layers. Moreover, it retains the entity type information[3] at an appropriate level of granularity without requiring explicit entity typing.

The *second* contribution of this paper is a **training-time causal intervention technique** for mitigating entity bias based on the proposed SCM (§3.2). Specifically, we identify entities that are likely to share similar predictive information with the given entity. During training, we perturb embedding of the given entity within a convex hull constructed by embeddings of similar entities. During inference, we represent the entity with the center of the convex hull. Taking advantage of the continuous nature of the embedding space, this intervention does not rely on models specifically trained on natural language to estimate the label distribution of unnatural text, nor does it sacrifice predictive entity or contextual information.

The *third* contribution of this paper is to transform the training-time intervention into **in-context intervention for black-box LLMs** whose parameters are inaccessible, and logits are uncalibrated (§3.3). A significant advantage of the proposed SCM is that the causal intervention is carried out at the input layer, enabling its implementation within an in-context setting. Specifically, we replace entities with placeholders and define each placeholder

---

[3]Entity type information plays a crucial role in entity-driven tasks. For example, without knowing a more specific location type, it is impossible to differentiate between relations *born_in_city* and *born_in_country*.

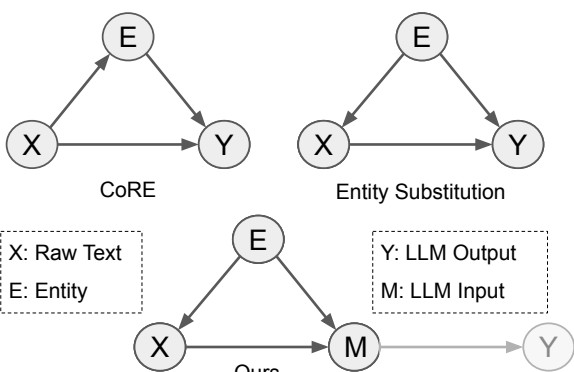

Figure 2: Structured causal models revealing entity bias.

by examples – a set of similar entities. For example, we can replace *Bill Gates* in Fig. 1 with *subject_entity* and prepend the prompt, *"Assume that subject_entity can be any of Steve Jobs, Bill Gates, and Jeff Bezos"*, to the input. This in-context intervention can be applied to any black-box LLM without additional cost.

Experiments on relation extraction (RE) and machine reading comprehension (MRC) show that the proposed causal intervention techniques are effective for both white-box and black-box LLMs. Under the white-box setting (§4), our training-time intervention significantly improves out-of-distribution performance of RoBERTa (Liu et al., 2019) on RE by 5.7 points and SpanBERT (Joshi et al., 2020) on MRC by 9.1 points, comparing with the vanilla version. Under the black-box setting (§5), our in-context intervention effectively reduces the entity-based knowledge conflicts (Longpre et al., 2021) and improves the task performance of GPT-3.5.[4] Specifically, our method outperforms the best baseline by up to 20.5 points of exact match accuracy on MRC and reduces the memorization ratio by up to 17.6 points on RE. Further analyses reveal the crucial role of the number of neighboring entities $k$ in balancing the predictive information and biasing information from entities, and the necessity of entity placeholder definition for in-context intervention.

## 2  Related Work

**Entity Bias in LLMs.** LLMs memorize factual knowledge in their parameters during pretraining (Roberts et al., 2020; Jiang et al., 2020) and show promising results in answering factual questions (Petroni et al., 2019; Brown et al., 2020; Wei

---

[4]https://platform.openai.com/docs/models/gpt-3-5

et al., 2022). However, the parametric knowledge may be inaccurate due to the misinformation in the training corpus (Lin et al., 2022) or outdated as the world evolves (Liska et al., 2022; Kasai et al., 2022). In such scenarios, it is critical for LLMs to update their predictions when provided with contextual evidence. However, previous studies (Longpre et al., 2021; Qian et al., 2021b; Yan et al., 2022) observe that language models may take entities as shortcuts, leading to spurious predictions based solely on parametric knowledge. This bias becomes more prominent when the evidence contains infrequent or conflicting knowledge compared to the training corpus.

To mitigate this bias, previous work (Longpre et al., 2021; Chen et al., 2022; Li et al., 2022; Zhou et al., 2023) introduces the entity substitution technique, which involves constructing counterfactual data by randomly replacing the entities, and updating the language models either by finetuning or in-context learning. Although showing improved results, these techniques are empirical and lack theoretical backgrounds. In this paper, we theoretically analyze the entity bias problem from a causal view. Furthermore, we propose a causal intervention method that surpasses the performance of entity substitution.

**Debiasing with Causal Intervention.** LLMs have been revealed with bias problems, for which literature has paid much attention in order to mitigate their adverse effects (Sweeney and Najafian, 2019; Zhang et al., 2020b; Venkit and Wilson, 2021; Lalor et al., 2022). Recent debiasing techniques incorporate the concept of counterfactual inference, and have been applied in various tasks for bias mitigation (Niu and Zhang, 2021; Qian et al., 2021a; Wang et al., 2022). One dominant technique is based on causal mediation analysis (Udomcharoenchaikit et al., 2022), which involves decomposing the total effect into pure direct effect and total indirect effect. In this context, Wang et al. (2022) utilize total direct effect and total effect to debias the relation extraction. Apart from debiasing, causal mediation analysis can be used to analyze biases in LLMs (Vig et al., 2020; Finlayson et al., 2021).

In addition to intervening causal mediator, previous studies have also explored confounder analysis (Keith et al., 2020; Qian et al., 2021a; Feder et al., 2022; Weld et al., 2022). A confounder is a variable that influences both the input and the output, causing a spurious correlation between them.

Typically, the de-confounder process applies the *do*-calculus (Pearl, 2012) to compute the prediction assuming that the value of the confounder variable is not the observed one but follows its natural distribution (Zhang et al., 2020a; Tian et al., 2022). Our approach is also based on confounder analysis. While nearly all the aforementioned approaches request a white-box accessibility of the model with at least logits of predictions, this work represents a pilot study of deconfounder method that applies to purely black-box LLMs.

## 3 Method

In this section, we first analyze methods for mitigating entity bias in a causal view and propose an easy-to-estimate SCM as a theoretical basis (§3.1). Based on the proposed SCM, we design a training-time intervention technique for white-box LLMs (§3.2) and an in-context intervention technique for black-box LLMs (§3.3).

### 3.1 Causal Analysis of Entity Bias

To compare existing methods in the same context, we analyze the structured causal models (SCMs) behind them. Fig. 2 shows two typical SCMs for entity bias mitigation methods, where $X$ refers to the raw input, $E$ refers to entities, and $Y$ refers to the label. The links $X \rightarrow Y \leftarrow E$ show that LLMs rely on both predictive information from the whole input and the biasing information from specific entities to make the prediction. The links $E \rightarrow X$ and $X \rightarrow E$ assume that the context is written down with the entity in mind or vice versa. As discussed by Verma and Pearl (1990), we cannot differentiate between these two directions merely based on statistical observations. Indeed, the two SCMs with opposite links between $X$ and $E$ are equivalent according to the Bayes' theorem:

$$
\begin{aligned}
&P(X)P(E|X)P(Y|X,E) \\
=&P(Y,X,E) \\
=&P(E)P(X|E)P(Y|X,E)
\end{aligned}
$$

As revealed by these SCMs, entity bias exists in LLMs because entities serve as either confounders or mediators. Thus, the bias can be mitigated through causal intervention, such as backdoor adjustment

$$
P(Y|do(X)) = \sum_{E} P(Y|X,E)P(E),
$$

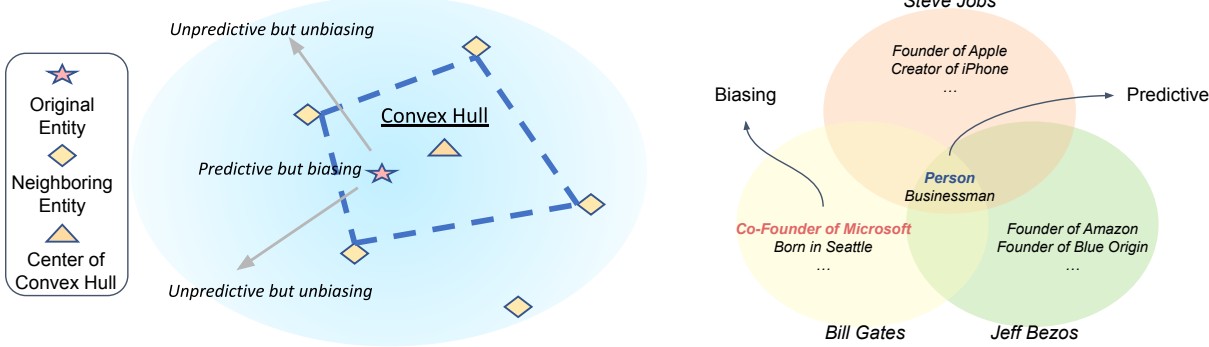

Figure 3: Left: Training-time intervention with $k = 4$. Right: Example of predictive and biasing information.

which eliminates the influence of a specific variable (in this context, $E$) by assigning values to this variable. However, previous SCM-based debiasing methods exhibit divergent performances, since they estimate different (conditional) probabilities using different surrogates when performing the causal intervention. For example, counterfactual analysis by Wang et al. (2022) estimates and deducts the biasing effect of entities on labels by masking the context, while Zhang et al. (2017) and Longpre et al. (2021) directly remove the effect of entities by entity masking or substitution. None of them estimates the causal effects of entity names precisely, due to the highly complex architectures of LLMs, which account for their unsatisfactory performance on mitigating entity bias.

In this work, we consider the SCM in Fig. 2, whose parameters are much easier to estimate in practice. Since most LLMs follow a sequential structure by stacking neural layers, mitigating the entity bias in one layer will also mitigate the entity bias in subsequent layers. The underlying logic is simple – if we block the spurious features in the input, there will be no spurious correlations to capture. Therefore, we propose to mitigate the entity bias in the input layer $M$, which could be an embedding layer or a prompt layer. Obviously, $P(M|X, E)$ can be estimated more accurately and efficiently than $P(Y|X, E)$, because there is no need to run the whole model, ensuring less error propagation and computational cost. To further improve the estimation by retaining as much predictive information as possible, we propose to estimate $P(M|do(X))$ by perturbing the entity with similar entities rather than masking it. In the following sections, we will show how to realize the proposed causal intervention on both white-box and black-box LLMs.

## 3.2 Training-time Intervention

For white-box models of which the parameters are accessible, we can effectively address their internal bias through training-time intervention. In the case of entity bias identified by the proposed SCM, we realize the causal intervention by perturbing the input entities or entity tokens using their neighboring counterparts in the embedding space, as shown in Fig. 3 (Left). For each entity presented in the input text, we first find its top $k$ nearest neighbors according to embedding distance. Then we construct the smallest convex hull[5] to cover the original entity and neighboring entities. Due to the continuous nature of the embedding space, the embeddings within the convex hull approximately represent the same predictive information as a whole. The entity-specific biasing information, which has the potential to trigger spurious shortcuts, gradually diminishes from the original entity towards the border of the convex hull.

During training, we introduce perturbations to the entity embedding by replacing it with a random embedding selected from within the convex hull. In this way, the convex hull bounded the predictive information, while random sampling further introduces noises and increases the diversity of data for robust training. During inference, we replace the original entity embedding with the center of the convex hull, in order to balance the trade-off between predictive and biasing information. Fig. 3 (Right) provides an example of the information preserved through such intervention. By replacing the entity *Bill Gates* with the center of the convex hull, encompassed by its neighboring entities, such as *Steve Jobs* and *Jeff Bezos*, we effectively retain the

---

[5]This convex hull-bounded perturbation is inspired by Dong et al. (2021), where perturbation within a convex hull formed by synonyms is used to improve model robustness against word substitutions.

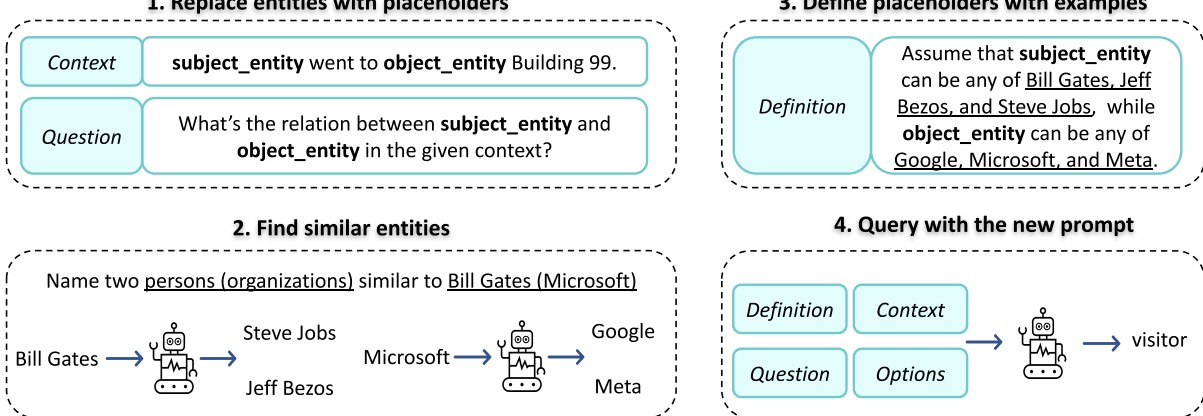

Figure 4: In-context intervention for black-box LLMs. We take relation extraction as an example.

shared predictive information (e.g., person), while mitigating the biasing information (e.g., founder of Microsoft). That is to say, the convex hull-bounded perturbation serves as an effective estimation of $P(M|do(X))$.

## 3.3 In-context Intervention

The rise of Web services powered by black-box LLMs, such as GPT-3.5, introduces new challenges for mitigating entity bias, demanding debiasing methods that do not require accessible model weights and prediction logits. As discussed in §3.1, a key advantage of our SCM is that the deconfounder operation is merely on the input layer. In the context of black-box LLMs, the input is the user-provided prompt. Thus, we perform the causal intervention solely through modifying prompts to resolve entity bias. We propose a four-step (test-time) in-context intervention technique for black-box LLMs. Fig. 4 shows the whole process.

First, we replace the original entity mention in the input with abstract placeholders (e.g., [ENTITY]). This step effectively mitigates any biasing information from the original entity names, because the placeholders are semantic-neutral. However, this step also eliminates predictive information from entities. We show in §5.3 that, without proper definition for the placeholder, models can easily fail to answer questions. In the next two steps, we construct definitions to provide predictive information for each placeholder while introducing minimal additional biasing information. Second, we query the LLM to name $k$ entities similar to the

original one (e.g., $E_o$).[6] These generated entities (e.g., $E_a$ and $E_b$) present similar predictive information as the original entity, and are able to fulfill the same function as neighboring entities in §3.2. Third, we define the placeholder with the original entity and generated entities. For example, we can verbalize the definition as "Assume [ENTITY] can be any of $E_o$, $E_a$ and $E_b$". This definition encourages the LLM to find common properties of given entities rather than relying on biasing information of one specific entity. The resulting placeholder along with its definition serves as an effective estimation of $P(M|do(X))$. Finally, we prepend the placeholder definition to the modified context and question, and query the LLM with the new prompt. This four-step adjustment ensures that the resulting prompt is free of specific biasing information pertaining to the original entity while still preserving sufficient predictive information by considering given entity examples as a whole.

## 4 White-Box Experiments

In this section, we evaluate our training-time intervention under the white-box setting.

### 4.1 Experimental Setup

**Datasets and Metrics.** We evaluate our methods on relation extraction (RE) and machine reading comprehension (MRC). For both tasks, we fine-tune models on an in-distribution (ID) training set and evaluate models on both ID and out-of-distribution (OOD) test sets. For RE, we adopt TACRED (Zhang et al., 2017) as the ID dataset and

---

[6]Here, we rely on the entity knowledge possessed by LLMs. However, it is possible to replace the LLM with external databases or tools in this step.

| | RE (F1) | | | MRC (EM) | | |
|---|---|---|---|---|---|---|
| | ID | OOD | $\Delta$ | ID | OOD | $\Delta$ |
| Vanilla Model | $71.1_{\pm 0.9}$ | $62.3_{\pm 0.6}$ | $-12.4\%$ | $79.1^{\dagger}_{\pm 0.1}$ | $63.1^{\dagger}_{\pm 0.8}$ | $-20.2\%$ |
| + Continual Pretraining (Yan et al., 2022)* | - | - | - | $\mathbf{79.6}^{\dagger}_{\pm 0.6}$ | $65.9^{\dagger}_{\pm 1.1}$ | $-17.2\%$ |
| + CoRE (Wang et al., 2022) | $\mathbf{71.3}_{\pm 0.3}$ | $61.2_{\pm 0.6}$ | $-14.2\%$ | - | - | - |
| + Entity Mask (Zhang et al., 2017) | $61.4_{\pm 0.5}$ | $61.9_{\pm 0.5}$ | $+0.9\%$ | $75.7_{\pm 0.6}$ | $62.9_{\pm 0.4}$ | $-16.9\%$ |
| + Entity Substitution (Longpre et al., 2021) | $66.6_{\pm 0.6}$ | $65.8_{\pm 0.3}$ | $-1.2\%$ | $76.4_{\pm 0.8}$ | $70.8_{\pm 1.5}$ | $-7.3\%$ |
| + Ours | $70.8_{\pm 0.3}$ | $\mathbf{68.0}_{\pm 0.3}$ | $-3.9\%$ | $77.0_{\pm 0.7}$ | $\mathbf{72.2}_{\pm 0.5}$ | $-6.2\%$ |

Table 1: Results under white-box setting. We report the average F1/EM score and standard deviation of three runs. $\Delta$ shows the relative performance change between ID and OOD. The best number of each column is in bold. * Continual pretraining is not directly comparable to finetuning methods. [†] Numbers copied from Yan et al. (2022).

EntRED (Wang et al., 2023) as the OOD dataset, and report micro-F1 score. In both datasets, entities in each sentence are given. For MRC, we adopt TriviaQA (Joshi et al., 2017) as the ID dataset and its answer-substituted version (Yan et al., 2022) as the OOD dataset, and report exact match (EM) score. Following Yan et al. (2022), we hold out 10% of the training data for development and evaluate models on the original development set. We use the DBName version of their OOD dataset. For all metrics, we report the average score with standard deviation of three runs.

**Baselines.** We compare our methods with the following baselines. *Entity Mask* (Zhang et al., 2017) masks the subject and object entities in the sentence with special tokens. *Entity Substitution* (Longpre et al., 2021) randomly selects an entity of the same type to substitute the original entity. *CoRE* (Wang et al., 2022) applies counterfactual inference by computing the difference between the prediction made with the entire sentence and the prediction made with only the entities observed. *Continual Pretraining* (Yan et al., 2022) introduces an intermediate pretraining stage to the backbone model with the objective of recovering masked entities.

**Implementation Details.** For RE, we apply RoBERTa (Liu et al., 2019) as the backbone model following previous works (Zhou and Chen, 2022; Wang et al., 2022). We use the `entity_marker_punct` input format from Zhou and Chen (2022) in main experiments, in order to mitigate the impact of explicit entity type information on our analysis of entity bias. For MRC, we apply SpanBERT (Joshi et al., 2020) as the backbone model following Yan et al. (2022). Since entities are not given in MRC datasets, we use the same named entity recognition tool used by Yan et al. to

extract entities. Since the detected entities could be noisy and incomplete, we perform our method upon answer-substituted training set ensuring all answer entities are perturbed as strong as *Entity Substitution*. Since RoBERTa and SpanBERT lack entity-level embeddings, we apply our causal intervention to each token embedding within the entity mention instead. To construct convex hull, We select neighboring tokens based on their Euclidean distance to the original token in the embedding space. For both tasks, we perform training-time intervention on each entity token with $k = 3$. While further data augmentation is always possible, for a fair comparison, we finetune all the models with the same amount of data. More implementation details are in Appx. §A.1.

### 4.2 Results

As shown in Tab. 1, the vanilla RoBERTa and SpanBERT experiences significant declines in performance on RE (-12.4%) and MRC (-20.2%) when evaluated on OOD test sets. For both tasks, the OOD test set exhibits lower entity bias, achieving better performance on it suggests that the model relies less on entity bias as a predictive factor.

*CoRE* and *Continual Pretraining* are the only baselines that improve the ID performance. CoRE leads to a slight performance decrease on the OOD test set of RE in exchange,[7] while *Continual Pretraining* further increases the OOD performance on MRC. *Entity Mask* successfully narrow down or even reverse the relative performance drop under OOD setting on the two tasks. However, its absolute performance decreases significantly due

---

[7]This is because *CoRE* is designed for a class-balanced setting, but this experiment emphasizes the performance on the raw class distribution. Moreover, we search its bias mitigation weight on the ID development set, which has a notably different entity distribution compared with the OOD test set.

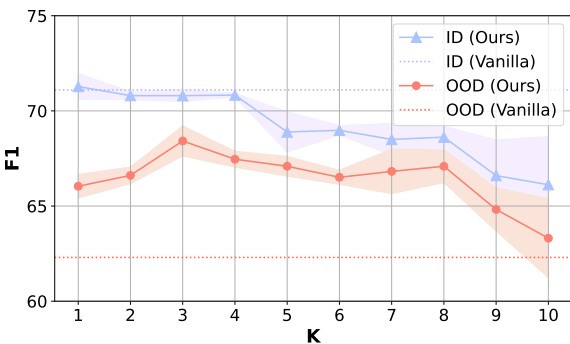

Figure 5: F1 score of training-time intervention with different $k$ on RE.

to the loss of predictive information from entities. Moreover, its effectiveness is dependent on the task property. Unlike MRC, entities are given and are not answers in RE, so the gap between ID and OOD performance of *Entity Mask* are much smaller. *Entity Substitution* stands out among all the baselines in terms of the OOD performance, with an absolute improvement of 3.5 points on RE and 7.7 points on MRC. However, its ID performance suffers a lot from the distribution shift of entities during training.

Our training-time intervention achieves the best OOD performance, with an absolute improvement of 2.2 points on RE and 1.4 points on MRC compared with *Entity Substitution*. At the same time, its ID performance is also better. These results show that our method mitigates entity bias more effectively without losing much predictive information. In other words, the proposed method represents a better way to estimate the parameters of the proposed SCM accurately.

### 4.3 Analysis

To provide a comprehensive understanding of our training-time intervention, we further conduct analyses on RE.

**Effect of $k$.** The number of neighbors, $k$, plays a crucial role in balancing the predictive information and biasing information from entities. To find the sweet spot of $k$, we examine its influence on model performance as shown in Fig. 5. In general, the ID performance decreases when $k$ increases. As the value of $k$ increases, the resulting convex hull becomes larger, causing the center of the hull to move further away from the original entity. Consequently, both the predictive information and biasing information that contribute to ID performance gradually diminish. In contrast, the OOD performance is lower when $k$ is too big or too small. When $k$ is too big, the same problem under ID setting also happens to the OOD setting. When $k$ is too small, the biasing information is not effectively mitigated, because the perturbed entity is too close to the original entity.

**Entity Type as Input.** Previous experiments in this section do not explicitly input entity information as it may disturb the causal analysis. Here, we analyze the effect of entity type information as input. We use the `typed_entity_marker_punct` input format from Zhou and Chen (2022). The ID and OOD F1 scores of vanilla RoBERTa model are 74.6 and 68.9 points, respectively. Our training-time intervention further improves the ID performance by 0.7 points and the OOD performance by 2.9 points. These results indicate that information from neighboring entities is complementary to coarse-grained entity type information for precise RE.

## 5 Black-Box Experiments

In this section, we evaluate our in-context intervention for mitigating entity bias from LLMs under black-box setting.

### 5.1 Experimental Setup

**Datasets.** Following Zhou et al. (2023), we adopt GPT-3.5 `text-davinci-003` as the backbone LLM and evaluate the model performance under a zero-shot setting. We use the RE and MRC datasets provided by Zhou et al. (2023). The RE dataset is based on Re-TACRED (Stoica et al., 2021). Zhou et al. pair each instance's entities with a randomly sampled context that shares the same entity types but possesses different relations. To mitigate the influence of the label *no_relation*, which can also serve as a signal of abstention, we further filter out all instances whose original or updated labels are *no_relation*. The MRC dataset is based on Natural Questions (Kwiatkowski et al., 2019). Zhou et al. replace the original answer in each instance with a randomly sampled entity of the same type. They only collect instances where the LLM can give the correct answer based on the raw context. Intuitively, LLMs that faithfully capture contextual information should update their answers based on the new context.

**Metrics.** We report the F1 score for RE, and EM score for MRC. To align with previous works, we

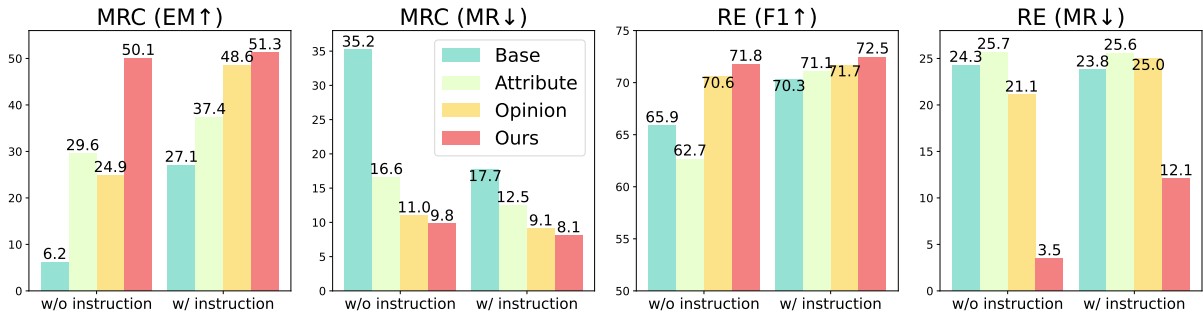

Figure 6: GPT-3.5 results on MRC and RE under black-box setting. We report the EM score on MRC and the F1 score on RE, for which higher scores are better. We also report the MR score on both tasks, for which lower scores are better. Our in-context intervention performs consistently better than baselines under all settings.

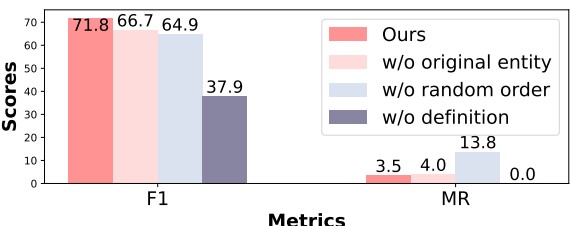

Figure 7: Ablation study of in-context intervention for GPT-3.5 on RE.

also report the memorization ratio (MR; Longpre et al. 2021) to measure the model's ability to update answers based on given contexts.[8]

**Baselines.** We compare our in-context intervention with the methods introduced by Zhou et al. (2023). *Base* prompts directly concatenate the context and the question of each instance as the query. *Attribute*-based prompts append *"in the given context"* to the question. *Opinion*-based prompts modified the context to a narrator's statement by prepending *"Bob said"* to the context, and then query the LLM about the narrator's opinion by prepending *"What's Bob's opinion on"* to the question. We evaluate all methods with and without specifically designed task instructions following Zhou et al. (2023).

**Implementation Details.** We apply our in-context intervention to attribute-based prompts. We adopt the backbone LLM to propose two similar entities along with the original entity to define each placeholder. To further eliminate the spurious entity mapping, we shuffle the entities for each placeholder before verbalization. Details of all prompt templates used can be found in Appx. §A.2. Since

entities are not given in MRC, we detect named entities and replace them with placeholders using `gpt-3.5-turbo` as an external tool. Given the potential abundance of entities in long contexts, we do not replace entities that exclusively appear in the context.

### 5.2 Results

As shown in Fig. 6, all methods benefit from carefully designed task instructions in terms of task performance. The *Opinion*-based prompt performs the best among all baselines in most cases. Compared with the *Base* prompt, it significantly improves the EM score by 18.7-21.5 points on MRC and the F1 score by 0.6-4.7 points on RE. Our in-context intervention achieves the highest EM/F1 score and the lowest MR score under all settings. Specifically, without task instruction, our in-context intervention outperforms the best baseline by 20.5 EM points on MRC and reduces the MR score by 17.6 points on RE. These results demonstrate the effectiveness of our causal intervention for addressing entity-based knowledge conflicts in black-box LLMs.

### 5.3 Ablation Study

We in addition conduct an ablation study on RE to provide a comprehensive understanding of our method, as shown in Fig. 7. When the placeholder definition is not provided (i.e., *w/o definition*), no entity information, including both biasing and predictive information, appears in the input. As a result, it successfully blocks any spurious shortcuts with MR drops to 0. However, the F1 score also drops sharply from 71.8 points to 37.9 points, indicating that some entity information is essential to accurate RE and the LLM cannot understand the placeholders well without their definition.

---

[8] $MR = \frac{P_o}{P_o + P_s}$, where $P_o$ is the probability that the model generates the original answer and $P_s$ is the probability that the model updates the answer correctly.

We further examine the role of original entities in the placeholder definition. On the one hand, we remove the original entities from the definition (i.e., *w/o original entity*). Results show that our method can still improve F1 while reducing MR. This verifies the effectiveness of using a set of similar entities to represent the predictive information from the original entity. On the other hand, we put the original subject and object entities at the same position (i.e., *w/o entity shuffle*) in the definition so that the LLM can easily map them. As a result, the MR increases significantly, showing that the LLM can find spurious shortcuts even through mapping the subject entity and the object entity from two entity sets.

## 6 Conclusion

In this paper, we analyze the entity bias in LLMs from a causal view. Building upon an SCM whose parameters are easier to estimate, we propose training-time causal intervention for white-box LLMs and in-context causal intervention for black-box LLMs. Both intervention techniques perturb the original entity with neighboring entities to mitigate spurious correlations between specific entities and predictions. Experiments on relation extraction and machine reading comprehension show that the proposed intervention can effectively reduce the conflicts between parametric knowledge and contextual knowledge and significantly improve the performance of LLMs. Future work can apply our causal intervention to more LLMs and tasks to achieve context-faithful answers.

## Acknowledgement

We appreciate the reviewers for their insightful comments and suggestions. Fei Wang is supported by the Annenberg Fellowship and the Amazon ML Fellowship. Wenjie Mo is supported by the USC CURVE Fellowship and the Provost's Research Fellowship. Wenxuan Zhou and Muhao Chen are supported by the NSF Grant IIS 2105329, the NSF Grant ITE 2333736, the DARPA MCS program under Contract No. N660011924033 with the United States Office Of Naval Research. This work is also supported in part by a Cisco Research Award, two Amazon Research Awards, and a Keston Research Award. Computing of this work has been partly supported by a subaward of NSF Cloudbank 1925001 through UCSD.

## Limitation

Although we have tried to verify the effectiveness of our method under diverse settings, including different LLMs, different accessibility of model parameters, and different tasks, there are always more options for further investigation, especially nowadays when more and more LLMs are kept produced. Considering the property of the entity bias issue may vary when it comes to different LLMs and datasets from different domains, future work can build better benchmark for more comprehensive evaluation. In this paper, we only consider zero-shot prompting for black-box LLMs, because this will help us to control variables during causal analysis. However, it is possible to combine the proposed causal intervention with cutting-edge LLM inference methods, such as in-context learning (Brown et al., 2020), although the underlying SCM may become more complex.

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

# A  Implementation Details

## A.1  White-Box Experiments

For RE, we use RoBERTa-Large as our backbone model, which has 354 million parameters. Our implementation is based on the codebase by Zhou and Chen (2022) with their default hyper-parameters. More specifically, we employ a learning rate of 3e-5, a batch size of 32, and conduct training for a total of 5 epochs. Other method-specific hyper-parameters are selected on the development set of TACRED. Finetuning typically takes 1.5 hours on an NVIDIA RTX A5000 GPU.

For MRC, we use SpanBERT-base-cased as our backbone model, which has 110 million parameters. Our implementation is based on the codebase by Yan et al. (2022) with their default hyper-parameters. More specifically, we employ a learning rate of 2e-5, a batch size of 16, and conduct training for a total of 4 epochs. Other method-specific hyper-parameters are selected on the hold-out development set of TriviaQA. Finetuning typically takes 3 hours on an NVIDIA RTX A5000 GPU.

## A.2  Black-Box Experiments

Our implementation is based on the codebase by Zhou et al. (2023).

The instruction for MRC is

> Instruction: read the given information and answer the corresponding question.

The prompt without instruction for MRC is

> Assume that {ENTITY0} can be any of {entity0_candidates}. [Assume that {ENTITY1} can be any of {entity1_candidates} ...]
> {context}
> Q:{question} based on the given text? Extract the answer from the given text. Do not add other words.
> A:

The instruction for RE is

> Identify the relationship between two entities from a list of options.

The prompt without instruction for RE is

> Assume that subject_entity is one of {subj_candidates}, while object_entity is one of {obj_candidates} in the following text.
> {context}
> Q: Which option indicates the relationship between subject_entity and object_entity in the given text?
> Options:{options}
> A:

The prompt template for detecting entities in MRC is

> List named entities in the following sentence. Separate the entities with ###, if you find multiple entities. Do not add additional words before or after your answers.
> {sentence}

The prompt template for replacing entities with placeholders in MRC is

> Replace the entity {entity_list} in the following paragraph.
> {paragraph}

The prompt template for finding similar entities is

> Name two [{entity_type}] entities similar to "{entity}". Separate the entities with ###, and do not add additional words before or after your answers. Provide random answers if you are not sure.

In all the above prompts, variables are surrounded with curly brackets and optional variables are surrounded with square brackets.