# OpenReview forum: "A Causal View of Entity Bias in (Large) Language Models"
_EMNLP/2023/Conference — EMNLP 2023 Findings_

### Official Review · Reviewer_zF98 · 2023-07-24

**Soundness:** 3

**Excitement:**

3: Ambivalent: It has merits (e.g., it reports state-of-the-art results, the idea is nice), but there are key weaknesses (e.g., it describes incremental work), and it can significantly benefit from another round of revision. However, I won't object to accepting it if my co-reviewers champion it.

**Paper Topic And Main Contributions:**

This paper proposes a structured causal model (SCM) to mitigate entity bias in PLMs/LLMs.  To mitigate the conflicts between parametric knowledge and contextual knowledge, the paper first replaces entities with placeholders for the input sentence and then finds similar entities via LLM, thirdly generates the definition of placeholders with similar entities, and finally combines the definition, context, and prompt question to a whole input sentence. The experimental results show that the framework proposed by this paper could get better results compared with baseline models.

**Reasons To Accept:**

The paper proposes a causal view of entity bias in PLM/LLM models, which is interesting and has practical significance.


**Reasons To Reject:**

Various prompt-based learning methods have should that the classification results made by PLM/LLM can be affected by prompt templates.
s it unclear whether the improvement in results from this method is specific because it mitigates the entity bias in PLM/LLM, or generates a more specific prompting template for PLM/LLM to make decisions.

I think that LLM does not have entity bias because it has been pre-trained on a large amount of unsupervised/semi-supervised data. The entity bias in prompts given to LLM may be the reason why there is entity bias in  LLM inference results. The paper should analyze this point of view.

 Using the top k nearest neighbors to construct the convex hull seems like the prototype-based method. As the value of k increases, the center point of the convex hull should theoretically be closer to the true meaning of this entity category. In other words, the center point of the box will have less bias towards this entity category. This means the center of the hull moves further away from the original entity. But it also means the original entity has a bigger bias, why the results become worse?


**Reproducibility:**

3: Could reproduce the results with some difficulty. The settings of parameters are underspecified or subjectively determined; the training/evaluation data are not widely available.

**Reviewer Confidence:**

3: Pretty sure, but there's a chance I missed something. Although I have a good feel for this area in general, I did not carefully check the paper's details, e.g., the math, experimental design, or novelty.

---

> ### Author Rebuttal · Authors · 2023-08-27
>
> We appreciate the valuable feedback from the reviewer.
>
> **Responses to reasons to reject**
>
> > Various prompt-based learning methods have should that the classification results made by PLM/LLM can be affected by prompt templates. s it unclear whether the improvement in results from this method is specific because it mitigates the entity bias in PLM/LLM, or generates a more specific prompting template for PLM/LLM to make decisions.
>
> We only perturb entities based on existing templates (lines 508-509). Hence, the proposed causal intervention only introduces minimal modifications to the prompt template. Taking RE as an example, to make a fair comparison, we only replace entities with placeholders and prepend definitions of placeholders. All other content in the prompt remains the same for all methods.
>
> > I think that LLM does not have entity bias because it has been pre-trained on a large amount of unsupervised/semi-supervised data. The entity bias in prompts given to LLM may be the reason why there is entity bias in LLM inference results. The paper should analyze this point of view.
>
> Previous (Longpre et al., 2021, Yan et al., 2022, Zhou et al., 2023) and our work have shown that PLMs/LLMs ranging from BERT to GPT3.5 all suffer from entity bias. The main reason for this is the unbalanced association between entities, and between entities and properties. For example, Bill Gates is much more often associated with Microsoft than most of other entities or concepts. Such biased association of concepts easily causes models to tend to pick up prediction shortcuts during training. During inference, entities in prompts will trigger shortcuts, making the model ignore the actual context. The problem is that LLMs are not faithful to the contextual knowledge, but overly rely on its parametric knowledge.
>
> > Using the top k nearest neighbors to construct the convex hull seems like the prototype-based method. As the value of k increases, the center point of the convex hull should theoretically be closer to the true meaning of this entity category. In other words, the center point of the box will have less bias towards this entity category. This means the center of the hull moves further away from the original entity. But it also means the original entity has a bigger bias, why the results become worse?
>
> When $k$ becomes extremely large (e.g., as large as the number of all entities or the size of the whole vocabulary), the center of the convex hull won’t necessarily be the entity category but an average of all entities or tokens. Model performance drops because both biasing and predictive information are lost. The tradeoff between unbiasing and predictive information can be balanced by $k$. A proper $k$ makes the center of the convex hull closer to the category of proper granularity. As a result, we observe that when $k$ increases, the OOD performance first increases and then decreases.

---

### Official Review · Reviewer_r268 · 2023-08-04

**Soundness:** 3

**Excitement:**

3: Ambivalent: It has merits (e.g., it reports state-of-the-art results, the idea is nice), but there are key weaknesses (e.g., it describes incremental work), and it can significantly benefit from another round of revision. However, I won't object to accepting it if my co-reviewers champion it.

**Paper Topic And Main Contributions:**

This paper introduces techniques for mitigating entity bias as both a training-time intervention and prompting intervention for white-box and black-box LLMs respectively. They conduct an analysis of past entity bias mitigating techniques, and introduce a structured casual model to frame their technique.  The causal method is separated into a training time intervention, which consists of swapping a specific entities embedding in the lowest layer with an embedding at the center of a convex hull built off of closely related entities.  They also propose a method for use with black-box LLMs, which focuses on prompting the LLM to consider related entities in addition to a specific one.

They experiment with their interventions on the two earlier mentioned styles of LLMs and show that the training-time intervention improves over past methods in this study, specifically in the OOD performance. They also show that their prompting intervention aids black-box LLMs in mitigating entity bias as compared to their baseline techniques.


**Questions For The Authors:**

A) Where do you get the set of entity neighbors to compute the convex hull from?  Were they extracted from the relevant datasets, or from an associated KB?

B) In figure 1 there is an option for the model to choose from two given options in order to answer the entity relation question, however, in figure 4 that option is not present. Are options used in your experiments, and if so, how does the method work without those given options?

(If no options given), One issue I see is that the method may not debias across multiple axes. How do you ensure that the training time intervention can debias across multiple axes? In the given example, Bill Gates is decoupled from his role as a founder and is shown to be a visitor. But how can the intervention debias across his nationality, his gender, etc.?

C) In in-context example learning, how can the prompting strategy be applied to multiple entities? Has any analysis been done on if entity bias is sought for only a fraction of the entities present in the sample and what effect that has on performance?

D) Can you elaborate more on why in-domain performance decreases when using your method? In what situations would a practitioner utilize this method given this drop?

E) For in-context learning, can’t entity bias be mitigated through other prompting techniques, such as demonstrations via in-context examples, CoT techniques, self-checking, etc.? How would these methods compare as a baseline against your method?

F) I would like further discussion of this point from the limitations section: “In this paper, we only consider zero-shot prompting for black-box LLMs, because this will help us to control variables during causal analysis.”


**Reasons To Accept:**

A) The paper is generally well-written and does a thorough analysis of past work in the entity bias space.
B) Figures 3 and 4 in particular are well-illustrative of the techniques used.
C) The ablation study on the in-context prompting section also shows clearly which components of their prompting strategy contribute to performance.
D) Methods that use causal approaches with LLMs are important in achieving high performance in applications and reducing bias/errors.
E) The authors report a robust set of experiments with multiple runs and error bars.

**Reasons To Reject:**

A) In-domain performance drops when applying the method as a training-time intervention. Although this is understandably an OOD mitigation technique, it would be nice to have consistent performance in domain
B) There could be additional experiments in the in-context prompting section, as some important baselines for in-context prompting have not been included.


**Reproducibility:**

3: Could reproduce the results with some difficulty. The settings of parameters are underspecified or subjectively determined; the training/evaluation data are not widely available.

**Reviewer Confidence:**

3: Pretty sure, but there's a chance I missed something. Although I have a good feel for this area in general, I did not carefully check the paper's details, e.g., the math, experimental design, or novelty.

**Typos Grammar Style And Presentation Improvements:**

The “PLM” usage in the abstract is not defined.

Figure 3 caption could use further explanation so that it is more easily understood on quick passes.

---

> ### Author Rebuttal · Authors · 2023-08-27
>
> We appreciate the valuable feedback from the reviewer.
>
> **Responses to reasons to reject**
>
> > A) In-domain performance drops when applying the method as a training-time intervention. Although this is understandably an OOD mitigation technique, it would be nice to have consistent performance in domain
>
> We appreciate the reviewer’s understanding that methods designed for the OOD setting tend to lead to ID performance drop. Under the ID setting, a biased model can overfit the data distribution, therefore not surprisingly performing better on the test set presenting the same types of bias. Specifically in our method, the tradeoff between OOD and ID performance can be adjusted by $k$. As shown in Fig 5, when $k=1$, our method improves OOD performance by 4.1 points of F1 *without ID performance drop*.
>
> > B) There could be additional experiments in the in-context prompting section, as some important baselines for in-context prompting have not been included.
>
> We did not compare our method with prompting techniques like CoT as they were not designed for addressing entity bias. In response to this concern, we extend the experiments to evaluate CoT on RE w/ instruction. In comparison to the base prompts, CoT does not lead to a performance increase in terms of F1, though reduces MR by 3.3 points. In contrast, our method increases the F1 score by 2.2 points and reduces MR by 11.7 points. We are happy to add this among the baseline results we have had.
>
> Adding examples to the prompt will make the structured causal model more complex. As a first attempt to provide a causal analysis of entity bias in black-box LLMs, we focus on the zero-shot setting. Future work may consider extending our framework to the few-shot setting.
>
> **Responses to questions**
>
> > A) Where do you get the set of entity neighbors to compute the convex hull from? Were they extracted from the relevant datasets, or from an associated KB?
>
> We do not rely on external resources to create the convex hull. Specifically, we find neighboring tokens of each entity token based on token-level embedding similarity (line 275 and footnote 7). This is a rough proxy of finding similar entities, but is shown to be sufficiently effective in our experiments. Using associated KB may be helpful, but we chose to keep it simple in this paper.
>
> > B) In figure 1 there is an option for the model to choose from two given options in order to answer the entity relation question, however, in figure 4 that option is not present. Are options used in your experiments, and if so, how does the method work without those given options?
>
> Following prior work (Zhou et al., 2023), we provide options for RE but not MRC under the black-box setting. We’ll update the figures accordingly. In MRC, we perform the causal intervention to entities in the questions.
>
> > (If no options given), One issue I see is that the method may not debias across multiple axes. How do you ensure that the training time intervention can debias across multiple axes? In the given example, Bill Gates is decoupled from his role as a founder and is shown to be a visitor. But how can the intervention debias across his nationality, his gender, etc.?
>
> Considering (multi-perspective) contextual similarity of entities could be interesting and is worthy of investigation by follow-up works. In this work, as a pilot study for debiasing both PLMs and black-box LLMs, we focused on a general setting of entity similarity though. A possible solution that future work may consider could be to decompose the entity embedding along multiple axes and create separate convex hulls based on the decomposed embedding.
>
> > C) In in-context example learning, how can the prompting strategy be applied to multiple entities? Has any analysis been done on if entity bias is sought for only a fraction of the entities present in the sample and what effect that has on performance?
>
> Our method does not rely on assumptions about the number and ratio of entities that trigger entity bias. For example, in MRC under the black-box setting, as we do not assume the awareness of which entity may cause the bias, we perform our in-context intervention on all entities in the question. This aligns with the condition that entity bias is “sought for a fraction of multiple entities”. As shown in Fig 6, our method is effective under this setting.
>
> > D) Can you elaborate more on why in-domain performance decreases when using your method? In what situations would a practitioner utilize this method given this drop?
>
> The ID performance drops because the debiasing method avoids overfitting the ID distribution (that presents the same types of bias). In practice, the tradeoff between OOD and ID performance can be adjusted by $k$, and $k$ can be selected based on the performance on development sets and user needs.
>
> > E) For in-context learning, can’t entity bias be mitigated through other prompting techniques, such as demonstrations via in-context examples, CoT techniques, self-checking, etc.? How would these methods compare as a baseline against your method?
>
> We did not compare our method with prompting techniques like CoT as they were not designed for addressing entity bias. In response to this concern, we extend the experiments to evaluate CoT on RE w/ instruction. In comparison to the base prompts, CoT does not lead to a performance increase in terms of F1, though reduces MR by 3.3 points. In contrast, our method increases the F1 score by 2.2 points and reduces MR by 11.7 points. We are happy to add this among the baseline results we have had.
>
> > F) I would like further discussion of this point from the limitations section: “In this paper, we only consider zero-shot prompting for black-box LLMs, because this will help us to control variables during causal analysis.”
>
> Adding examples to the prompt will make the structured causal model more complex. As a first attempt to provide a causal analysis of entity bias in black-box LLMs, we focus on the zero-shot setting. Future work may consider extending our framework to the few-shot setting.
>
>
> We also appreciate the editorial suggestions and will revise our paper accordingly.

---

### Official Review · Reviewer_wJyF · 2023-08-05

**Soundness:** 4

**Excitement:**

4: Strong: This paper deepens the understanding of some phenomenon or lowers the barriers to an existing research direction.

**Paper Topic And Main Contributions:**

The paper aims to eliminate entity bias in pre-trained language models through causal intervention methods. In particular, the paper proposes an entity perturbation method that perturbs the input text with similar entities from the convex hull of the target entity’s representation. Such causal intervention at the input layer is shown to be effective on both training-time and in-context interventions. The proposed debiasing approach is evaluated on relation extraction data sets (TACRED and EntRED) and reading comprehension data set (TriviaQA).  Experimental results suggest strong performance gain over other baselines.

**Questions For The Authors:**

1. How do you obtain the entity embeddings for constructing the convex hull during training time intervention? Are you using a pre-trained model that provides such embeddings?

2. How does debiasing impact long-tail entities? In particular, given the sparsity of the embedding space near the long-tail entities, how can we construct a convex hull?

3. What is the interpretation of OOD in the experiments? Are these the entities not seen in training?


**Reasons To Accept:**

1. Strong empirical results demonstrating the effectiveness of the approach.

2. The paper shows ingenuity in constructing a prompt that is effective for debiasing entity bias. It will be interesting to the community to follow up on this work and see whether similar approaches can help to mitigate other types of biases.

3. The paper is well written and easy to follow


**Reasons To Reject:**

1. It seems the effectiveness of the proposed method is over-reliant on either the LLMs ability to provide similar entities (in-context intervention) or a pre-trained model’s ability to retrieve similar entities (training-time intervention).

2. There are some missing details in the paper. See the “Questions” section for details.


**Reproducibility:**

4: Could mostly reproduce the results, but there may be some variation because of sample variance or minor variations in their interpretation of the protocol or method.

**Reviewer Confidence:**

2: Willing to defend my evaluation, but it is fairly likely that I missed some details, didn't understand some central points, or can't be sure about the novelty of the work.

**Typos Grammar Style And Presentation Improvements:**

Too many footnotes in the paper, some of these should be incorporated into the main text and some others should be placed in the Appendix section when they contain specific details, e.g., choice of an experiment set up.

---

> ### Author Rebuttal · Authors · 2023-08-27
>
> We appreciate the valuable feedback from the reviewer.
>
> **Responses to reasons to reject**
>
> > It seems the effectiveness of the proposed method is over-reliant on either the LLMs ability to provide similar entities (in-context intervention) or a pre-trained model’s ability to retrieve similar entities (training-time intervention).
>
> Collecting similar entities or entity tokens is a prerequisite of our method. As shown in our paper, it is easy to realize with either PLM-based similarity search or LLM generation. There could also be model-free alternatives, such as predefined entity clusters or any other similarity-based entity search.
>
> > There are some missing details in the paper. See the “Questions” section for details.
>
> We provide the details in response to each question below.
>
> **Responses to questions**
>
> > How do you obtain the entity embeddings for constructing the convex hull during training time intervention? Are you using a pre-trained model that provides such embeddings?
>
> For models without entity embeddings (such as RoBERTa used in this paper), we find neighboring tokens of each entity token based on token-level embedding similarity (line 275 and footnote 7). This is a rough proxy of finding similar entities, but is shown to be effective enough in our experiments.
>
> > How does debiasing impact long-tail entities? In particular, given the sparsity of the embedding space near the long-tail entities, how can we construct a convex hull?
>
> The impact on long-tail entities is an intriguing question. Although we haven’t looked close into it, we hypothesize that one possible problem could be the convex hull becoming significantly larger due to the sparse embedding neighborhood. In this case, we can set a threshold to cut off the perturbation. In other words, instead of the raw convex hull, another possible design choice could be using the overlap between the convex hull and a norm ball centering around the target embedding.
>
> > What is the interpretation of OOD in the experiments? Are these the entities not seen in training?
>
> The OOD datasets are built by substituting entities in a specific context to present counterfactual information so as to cut off the prediction shortcut between entities and labels (lines 365-370, 476-479, 484-486). For example, EntRED replaces the subject and object entities in a sentence with entities of the same type.
>
> We also appreciate the editorial suggestions and will revise our paper accordingly.

---

### Meta-Review · Area_Chair_dXBv · 2023-09-17

**Recommendation:** 3

**Metareview:**

This paper introduces an approach to mitigating entity bias in pre-trained language models by employing causal interventions structured around a specific causal model. These interventions replace original entities with neighboring ones to reduce biasing information while maintaining semantic relevance. Experimental results show significant improvements, especially in out-of-distribution performance across different datasets and modes of operation (white-box and black-box settings).

I agree with the reviewers that the approach to construct a prompt that is effective for debiasing models is likely of significance to future work in the field. The reviewers are generally of the view that the paper is methodically sound but may only appeal to a narrow audience. Their constructive feedback has highlighted areas that need further clarification in the paper as the authors have also appreciated in their responses.

Two reviewers were very appreciative of the experimental setup --- both the empirical results as well as the ablation studies. One reviewer had little to say on their reasons to accept but also pointed out that some of these issues could be specific to the prompts used, raising concerns with the experimental setup. The prompts shared in examples in the paper are fairly generic and likely what an average user would input so I don't see any issues with the experiment design as such, but it might be a worthy exploration to see whether such entity bias is prompt dependent or not.

Aside from the great suggestions offered in the reviews, I would also point the authors to Section 2.1 in [1] (which, to my knowledge, is one of the first papers to anonymize entities to prevent entity bias) which also follows a similar thought process to mitigate entity bias by modifying the corpora. Though there is not a 1:1 match between the two approaches, they are similar enough to warrant a discussion in this paper's related work.

[1] Hermann, K. M., Kocisky, T., Grefenstette, E., Espeholt, L., Kay, W., Suleyman, M., & Blunsom, P. (2015). Teaching machines to read and comprehend. Advances in neural information processing systems, 28.

---

### Decision · Program_Chairs · 2023-10-07

**Decision:**

Accept-Findings

**Comment:**

This paper introduces an approach to mitigating entity bias in pre-trained language models by employing causal interventions structured around a specific causal model. These interventions replace original entities with neighboring ones to reduce biasing information while maintaining semantic relevance. Experimental results show significant improvements, especially in out-of-distribution performance across different datasets and modes of operation (white-box and black-box settings).

I agree with the reviewers that the approach to construct a prompt that is effective for debiasing models is likely of significance to future work in the field. The reviewers are generally of the view that the paper is methodically sound but may only appeal to a narrow audience. Their constructive feedback has highlighted areas that need further clarification in the paper as the authors have also appreciated in their responses.

Two reviewers were very appreciative of the experimental setup --- both the empirical results as well as the ablation studies. One reviewer had little to say on their reasons to accept but also pointed out that some of these issues could be specific to the prompts used, raising concerns with the experimental setup. The prompts shared in examples in the paper are fairly generic and likely what an average user would input so I don't see any issues with the experiment design as such, but it might be a worthy exploration to see whether such entity bias is prompt dependent or not.

Aside from the great suggestions offered in the reviews, I would also point the authors to Section 2.1 in [1] (which, to my knowledge, is one of the first papers to anonymize entities to prevent entity bias) which also follows a similar thought process to mitigate entity bias by modifying the corpora. Though there is not a 1:1 match between the two approaches, they are similar enough to warrant a discussion in this paper's related work.

[1] Hermann, K. M., Kocisky, T., Grefenstette, E., Espeholt, L., Kay, W., Suleyman, M., & Blunsom, P. (2015). Teaching machines to read and comprehend. Advances in neural information processing systems, 28.